# Detecting the Critical States of Type 2 Diabetes Mellitus Based on Degree Matrix Network Entropy by Cross-Tissue Analysis

**DOI:** 10.3390/e24091249

**Published:** 2022-09-05

**Authors:** Yingke Yang, Zhuanghe Tian, Mengyao Song, Chenxin Ma, Zhenyang Ge, Peiluan Li

**Affiliations:** 1School of Mathematics and Statistics, Henan University of Science and Technology, Luoyang 471023, China; 2College of Agriculture, Henan University of Science and Technology, Luoyang 471023, China

**Keywords:** Type 2 diabetes mellitus (T2DM), dynamic network biomarker (DNB), critical state, sample specific network (SSN), network entropy, dark genes

## Abstract

Type 2 diabetes mellitus (T2DM) is a metabolic disease caused by multiple etiologies, the development of which can be divided into three states: normal state, critical state/pre-disease state, and disease state. To avoid irreversible development, it is important to detect the early warning signals before the onset of T2DM. However, detecting critical states of complex diseases based on high-throughput and strongly noisy data remains a challenging task. In this study, we developed a new method, i.e., degree matrix network entropy (DMNE), to detect the critical states of T2DM based on a sample-specific network (SSN). By applying the method to the datasets of three different tissues for experiments involving T2DM in rats, the critical states were detected, and the dynamic network biomarkers (DNBs) were successfully identified. Specifically, for liver and muscle, the critical transitions occur at 4 and 16 weeks. For adipose, the critical transition is at 8 weeks. In addition, we found some “dark genes” that did not exhibit differential expression but displayed sensitivity in terms of their DMNE score, which is closely related to the progression of T2DM. The information uncovered in our study not only provides further evidence regarding the molecular mechanisms of T2DM but may also assist in the development of strategies to prevent this disease.

## 1. Introduction

The development of many diseases, including T2DM, can be regarded as a nonlinear dynamic process which is generally divided into three states: normal state, pre-disease state/critical state, and disease state. The normal state is stable, characterized by stability and robustness, whereby any changes occur slowly. The disease state is a new stable state that represents a phase of deterioration in which more obvious signs and symptoms of the disease appear and many patients begin to receive treatment, but it is difficult to return to the normal state [1]. The pre-disease state/critical state corresponds to a critical point before the system transitions to an irreversible disease state and is accompanied by a drastic change in system dynamics [2]. In this state, the system is usually reversible, and reversion to the normal state is possible if there is appropriate treatment. Many diseases, including T2DM, are diagnosed upon reaching the disease stage. Therefore, it is important to identify the pre-disease state/critical state. However, this is a difficult task because there may be little difference between the normal and the pre-disease state/critical state to permit their clear distinction [3].

Traditional biomarkers can be used to distinguish disease states from normal states mainly based on the differential expression of individual molecules or a group of molecules, but such biomarkers cannot be used to detect the pre-disease state/critical state owing to their static nature [2]. In order to solve the challenge of detecting critical points, a new concept, called dynamic network biomarker (DNB), was proposed [3]. The expression of DNBs reflects the presence or severity of the disease state, and they are required to have constant values that are different in the respective disease and normal states such that they can be used as a signal for detection in the early stages of development of a complex disease. When a biological system in a normal state approaches the critical state, there are three statistical conditions that allow detection of a critical point based on DNBs: correlations between any two members of a DNB group rapidly increase, correlations between one member of a DNB group and any other non-DNB molecule rapidly decrease, and standard deviations of the variables in the DNB group drastically increase [4]. Based on these three necessary conditions, the DNB method can be used to identify a critical state. At present, research on combining the DNB method and information theory to detect the critical state of complex diseases is attracting attention. Based on the hidden Markov model (HMM), an inconsistent index algorithm was proposed to identify the critical state before disease deterioration [5]. Single-cell graph entropy (SGE) can explore the gene–gene associations among cell populations based on single-cell RNA sequencing data [6]. The single-sample-based Jensen–Shannon divergence (sJSD) method is used to detect the early-warning signals of complex diseases before critical transitions based on individual single-sample data [7]. The temporal network flow entropy (TNFE) method, which is based on network fluctuation of molecules, can detect the critical states of complex diseases on the basis of each individual [8].

In this study, we propose a new method, i.e., the degree matrix network entropy (DMNE) algorithm, based on sample-specific network (SSN) and network entropy. Dai et al. presented a new method to construct a cell-specific network (CSN) for each single cell, which transforms the data from “unstable” gene expression form to “stable” gene association form on a single-cell basis. This can also be directly applied to construct an individual network of each single sample [9]. Therefore, we applied this method to construct an SSN for the sample data and propose a method combining DMNE with network entropy that does not use single-cell data. Firstly, SSNs of reference samples and perturbed samples are constructed, respectively, and the network degree matrix (NDM) and local network are obtained. Then, the difference between the DMNE of the reference sample and the perturbed sample is calculated to quantify the differential network information flow. Finally, the DMNE score is calculated to characterize the molecular collective fluctuation or network fluctuation caused by case samples. By transforming floating gene expression datasets into stable network entropy, the DMNE method provides a new method to detect the critical state based on the SSN, which has the following advantages: (i) By calculating the DMNE score based on the difference of the network entropy at each stage, the DMNE method exhibits the differential changes in diseases at the network level. (ii) Based on the DMNE method, the critical states can be successfully detected, and the DNBs of the critical state can be effectively identified. (iii) Based on the DMNE method, “dark genes” are found, which play important roles during the development of T2DM, and drug targets against these are obtained.

## 2. Materials and Methods

### 2.1. Data Progression and Functional Analysis

The DMNE method was applied to detect the critical transitions during T2DM development and progression accompanying insulin resistance in adipose tissue, gastrocnemius muscle, and liver for rats (diabetes rats: Goto–Kakizaki (GK) rats; control rats: Wistar–Kyoto (WKY) rats). The high-throughput experimental datasets were downloaded from the NCBI GEO database (access ID: GSE13268, GSE13269, and GSE13270) (www.ncbi.nlm.nih.gov/geo (accessed on 1 October 2021)), comprising time-course gene expression data obtained from age-specific rats corresponding to well-designed time series (five samples each from 4, 8, 12, 16, and 20 weeks). These experimental datasets were provided by Almon et al. [10]. All of the GEO samples we used are provided in Table A1. In addition, pancreatic adenocarcinoma (PAAD) survival data and the gene expression values in different samples were downloaded from the cancer genome atlas (TCGA) database (https://cancergenome.nih.gov/ (accessed on 1 March 2022)).

The enrichment analysis of DNBs is based on Gene Ontology Consortium [11] (GOC, http://geneontology.org (accessed on 1 February 2022)), DAVID Bioinformatics Resources [12] (https://david.ncifcrf.gov/ (accessed on 1 February 2022)), and Circos [13] (http://www.circos.ca/ (accessed on 20 February 2022)). Protein–protein interaction (PPI) networks were drawn by STRING (https://string-db.org/ (accessed on 20 February 2022)) and the client software Cytoscape (https://cytoscape.org/ (accessed on 25 February 2022)). The drug targets were downloaded from DrugBank [14] (https://go.drugbank.com/ (accessed on 20 January 2022)), Therapeutic Target Database [15] (TTD, http://db.idrblab.net/ttd/ (accessed on 20 January 2022)), and Pharmacogenomics Knowledgebase [16] (PharmGKB, https://www.pharmgkb.org/ (accessed on 20 January 2022)).

### 2.2. Theoretical Background

DNBs form an observable subnetwork or molecular group in a disease system or disease network. They can be used to identify the critical state/pre-disease state before the sudden deterioration of complex diseases [17]. When the system approaches the critical state from the normal state, DNB molecules (i.e., genes or proteins) have the following three statistical properties [8]:The correlation between any two members of a DNB group rapidly increases;The correlation between one member in a DNB group and any other non-DNB molecule sharply decreases;The standard deviation for any member in the DNB group drastically increases.

These theoretical results have allowed many achievements in detecting critical points and in discussing related biological processes. According to these three properties, DNBs can be reliably identified, and significant early warning signals can be extracted.

The CSN method [9] provides a promising strategy for analyzing genes and gene associations at the single-cell level. This can also be applied to non-single-cell datasets for constructing an individual network of each single sample in a similar way [9]. We thus applied CSN to each sample to construct an SSN of reference samples (the healthy control samples) and perturbed samples (the diabetic samples), respectively. Therefore, network fluctuations can be quantified based on the differences. We propose a network entropy algorithm based on SSN to predict the critical state during the development of T2DM.

### 2.3. Data Preprocessing

We analyzed three preprocessed gene expression datasets measured in adipose, gastrocnemius muscle, and liver from diabetic rats (GK) and control rats (WKY) provided the same normal diet. The datasets were downloaded from the NCBI GEO database (access ID: GSE13268, GSE13269, and GSE13270). The three datasets were not raw data, and they were provided by Almon et al. [10]. The datasets comprised 31,099 probes, measured using the Affymetrix Microarray Suite 5.0 (Affymetrix), for 25 WKY controls and 25 spontaneously diabetic GK rats at 4, 8, 12, 16, and 20 weeks of age [10].

Then, we transformed the downloaded matrixes into required gene expression matrixes (GEMs) through ID conversion and the deletion of duplicate genes and null values. In the case of multiple probes corresponding to the same gene, the values were individually averaged for each to obtain three GEMs containing 15,246, 15,343, and 15,246 genes. Finally, the logarithm log(1 + *x*) was applied to normalize the three GEMs.

### 2.4. Algorithm to Detect the Tipping Point and Identify DNBs of T2DM Based on DMNE

After extracting some reference samples (samples from a normal cohort that are regarded as the background that represents the healthy or control individuals), we then developed and applied the following algorithm to detect the tipping point using the data of the diabetic case samples and healthy control samples. The algorithm flow is shown in Figure 1.

The steps of the degree matrix network entropy (DMNE) algorithm are specified as follows:

[Step 1] Construct the SSN of reference samples and perturbed samples, respectively.

Based on *n* reference samples $\left\{S_{1},S_{2},\ldots ,S_{n}\right\}$, the RSSN (the SSN of reference sample) is constructed as follows.

Firstly, we normalize the initial gene expression matrix with *m* rows/genes and *n* columns/samples and average the expression for the same gene in *n* samples. Secondly, we plot scatter diagrams for every two genes in a rectangular coordinate system, where the horizontal axis and the vertical axis are the expression value $E_{i}^{}$ of gene $g_{i}$ and the expression value $E_{j}^{}$ of gene $g_{j}$, respectively, and these two genes constitute a gene pair $\left(g_{i},g_{j}\right)$. *m* genes lead to *m*(*m* − 1)/2 scatter diagrams. Then, we draw two boxes near $E_{i}^{}$ and $E_{j}^{}$. According to the research of Dai et al. [9], the influence of box size and *p* value in different datasets on the clustering effect is tested from the viewpoint of clustering. It is indicated that the optimum box size is about 0.1, and the optimum *p* value is about 0.01 on average, which are set as the default parameters of the CSN method. The specific test results are given in Appendix A. We choose 0.1 *n*, which is proportional to the number of samples *n*, and we can adjust the parameters according to specific needs. The boxes are drawn according to a predetermined integer, and the two boxes overlap to obtain the third box. The value is obtained by calculating the number of points (samples) in the third box. Whether there is an edge between the two genes in the scatter diagram of genes $g_{i}$ and $g_{j}$ is determined by the following statistical dependence index:(1)$$r_{i,j}^{}=\frac{n^{}(E_{i},E_{j})}{n}-\frac{n^{}(E_{i})}{n}\frac{n^{}(E_{j})}{n}$$
where $n^{}(E_{i})$, $n^{}(E_{j})$ and $n^{}(E_{i},E_{j})$ represent the number of points (samples) in the vertical box, horizontal box, and overlapping box, respectively. If the statistical dependency index is greater than zero, there is an edge between $g_{i}$ and $g_{j}$; otherwise, there is no edge. That is, the RSSN is constructed based on the specific network of the reference samples.

Similarly, the case samples at different time points *t* = *T* are added to the reference samples to obtain the disturbed samples $\left\{S_{1},S_{2},\ldots ,S_{n},S_{case_{T}}\right\}$, and the $PSSN_{T}$ (the SSN of the perturbed samples at time point *T*) can be constructed.

[Step 2] Extract degree matrix and local network.

The NDM of reference samples and disturbed samples can be obtained from RSSN and $PSSN_{T}$, respectively.

Extract each reference local network $RLN^{d}(d=1,\ldots ,l)$ from RSSN. $RLN^{d}$ is composed of genes $g_{i}^{d}(i=1,2,\ldots ,M^{d})$ (the central gene is $g_{1}^{d}$, and the rest are neighborhood genes).

As the genes in the reference sample and the perturbed sample are the same, similarly, the perturbed local networks $PLN_{T}{}^{d}(d=1,\ldots ,l)$ are extracted from $PSSN_{T}$. The $PLN_{T}{}^{d}$ is composed of genes $g_{i}^{d}(i=1,2,\ldots ,M^{d})$ (the central gene is $g_{1}^{d}$, and the rest are neighborhood genes).

[Step 3] Calculate the node probability for every gene by fitting a Gaussian distribution.

Based on the NDM of reference samples, the Gaussian distributions of each gene $g_{i}^{d}(i=1,2,\ldots ,M^{d})$ in local network $RLN^{d}$ are fitted. Node probabilities $p_{reference}(x_{i}{}^{d})$ of gene $g_{i}^{d}$ are calculated as follows:(2)$$p_{reference}(x_{i}{}^{d})=\frac{\frac{1}{\sigma \sqrt{2\pi }}\int_{-\infty }^{x_{i}{}^{d}}e^{\frac{-{\left(u-\mu_{ir}{}^{d}\right)}^{2}}{2{\left(\sigma_{ir}{}^{d}\right)}^{2}}}du}{\sum_{i=1}^{M^{d}}\frac{1}{\sigma \sqrt{2\pi }}\int_{-\infty }^{x_{i}{}^{d}}e^{\frac{-{\left(u-\mu_{ir}{}^{d}\right)}^{2}}{2{\left(\sigma_{ir}{}^{d}\right)}^{2}}}du},$$
where $x_{i}{}^{d}$ is the degree value from NDM of gene $g_{i}^{d}$ in the reference sample, and $\mu_{ir}{}^{d}$ and $\sigma_{ir}{}^{d}$ are the mean value and standard deviation of degree value for gene $g_{i}^{d}$ in the reference samples, respectively.

Similarly, based on the NDM of perturbed samples, node probabilities $p_{perturbed}(x_{iT}{}^{d})$ of gene $g_{i}^{d}$ are, respectively, calculated as follows:(3)$$p_{perturbed}(x_{iT}{}^{d})=\frac{\frac{1}{\sigma \sqrt{2\pi }}\int_{-\infty }^{x_{iT}{}^{d}}e^{\frac{-{\left(u-\mu_{ip}{}^{d}\right)}^{2}}{2{\left(\sigma_{ip}{}^{d}\right)}^{2}}}du}{\sum_{i=1}^{M^{d}}\frac{1}{\sigma \sqrt{2\pi }}\int_{-\infty }^{x_{iT}{}^{d}}e^{\frac{-{\left(u-\mu_{ip}{}^{d}\right)}^{2}}{2{\left(\sigma_{ip}{}^{d}\right)}^{2}}}du},$$
where $x_{iT}{}^{d}$ is the degree value from NDM of gene $g_{i}^{d}$ in the perturbed sample at time point *t* = *T*, and $\mu_{ip}{}^{d}$ and $\sigma_{ip}{}^{d}$ are the mean value and standard deviation of the degree value for gene $g_{i}^{d}$ in perturbed samples at time *t* = *T*, respectively.

[Step 4] Calculate the network entropy of RSSN and $PSSN_{T}$.

Firstly, the network entropy of $RLN^{d}(d=1,\ldots ,l)$ is calculated as follows:(4)$$H^{n}(x_{i}{}^{d})=\frac{-\sum_{i=1}^{M^{d}}x_{i}{}^{d}p_{reference}(x_{i}{}^{d})\log(x_{i}{}^{d}p_{reference}(x_{i}{}^{d}))}{M^{d}}.$$

Then, the network entropy of the global network of RSSN can be calculated as
(5)$$H^{n}=\frac{\sum_{d=1}^{l}H^{n}(x_{i}{}^{d})}{l}.$$

The network entropy of $PLN_{T}{}^{d}(d=1,\ldots ,l)$ is
(6)$$H_{T}{}^{n+1}(x_{iT}{}^{d})=\frac{-\sum_{i=1}^{M^{d}}x_{iT}{}^{d}p_{perturbed}(x_{iT}{}^{d})\log(x_{iT}{}^{d}p_{perturbed}(x_{iT}{}^{d}))}{M^{d}}.$$

Then, the network entropy of the global network of $PSSN_{T}$ is
(7)$$H_{T}{}^{n+1}=\frac{\sum_{d=1}^{l}H^{n}(x_{iT}{}^{d})}{l}.$$

[Step 5] Calculate the DMNE score to quantify the network differences caused by the perturbed sample.

The DMNE of reference samples and perturbed samples is calculated as follows:(8)$$\Delta H_{T}=\left|H_{T}{}^{n+1}-H^{n}\right|$$
where the $\Delta H_{T}$ score reflects the global perturbation caused by the case samples at each time point *t* = *T*. The higher the score, the greater the difference between the case samples and the reference samples. The sudden increase in $\Delta H_{T}$ can be considered as an early warning signal of important changes in the process of disease progression.

## 3. Results

### 3.1. Detecting the Critical State of T2DM

Here, we apply the DMNE method to detect the critical transitions during T2DM development and progression accompanying insulin resistance in rat adipose, gastrocnemius muscle, and liver corresponding to the GSE13268, GSE13269, and GSE13270 datasets, respectively, which each contain 25 reference and 25 case samples. In the case samples, there are five samples at each stage. At each stage, the DMNE score is calculated. The top 200 genes (1.5% of all expressed genes) with the highest DMNE score at the critical state are regarded as DNBs.

#### 3.1.1. The Critical State of GSE13268

As shown in Figure 2a, the dramatic increase in the DMNE score for GK rat adipose appeared at 8 weeks, which indicates the upcoming critical transition. To show the DMNE scores in a local view, the landscape of the DMNE scores of DNBs is illustrated in Figure 2d. It can be seen that around 8 weeks, there is a group of genes whose DMNE scores abruptly increase. This critical phenomenon results from the drastic increase in the correlations between molecules in this group when the system approaches the tipping point. In Figure 2g, we illustrate the evolution of the top DMNE gene group/module, i.e., the protein–protein interaction (PPI) network of DNBs. This figure shows that a significant change in the network structure occurs at 8 weeks, signaling the critical transition into disease state from the molecular network level. The DNBs are given in Appendix A.

#### 3.1.2. The Critical State of GSE13269

As shown in Figure 2b, two critical transitions are detected for muscle tissue during T2DM. The dramatic increases in the DMNE score for GK rats appeared at 4 and 16 weeks, which indicate the upcoming critical transitions. To show the DMNE scores in a local view, the DMNE scores of DNBs are illustrated in Figure 2e. This figure shows that around 4 and 16 weeks, there is a group of genes whose DMNE scores abruptly increase. This critical phenomenon results from the drastic increase in the correlations between molecules in this group when the system approaches the tipping point. In Figure 2h, we illustrate the evolution of the top DMNE gene group/module, i.e., protein–protein interaction (PPI) network of DNBs. This figure shows that the significant changes in the network structure occur at 4 and 16 weeks, signaling the critical transitions into disease state from the molecular network level. The DNBs are given in Appendix A.

#### 3.1.3. The Critical State of GSE13270

As shown in Figure 2c, there are also two critical transitions during T2DM consistent with GSE13269. The dramatic increases in the DMNE score for GK rat livers appeared at 4 and 16 weeks, which indicate the upcoming critical transitions. To show the DMNE scores in a local view, the DMNE scores of DNBs are illustrated in Figure 2f. It can be seen that around 4 and 16 weeks, there is a group of genes whose DMNE scores abruptly increase. This critical phenomenon results from the drastic increase in the correlations between molecules in this group when the system approaches the tipping point. In Figure 2i, we illustrate the evolution of the top DMNE gene group/module, i.e., the protein–protein interaction (PPI) network of DNBs. This figure shows that the significant changes in the network structure occur at 4 and 16 weeks, signaling the critical transitions into disease state from the molecular network level. The DNBs are given in Appendix A.

### 3.2. Tissue-Specific Analysis

To confirm the significant relation between tissue-specific DNBs and T2DM progression, the GO analysis and KEGG enrichments of DNBs are conducted for each DNB to categorize the genes participating in different biological functions or pathways, as shown in Table 1, Table 2 and Table 3 and Figure 3. In addition, “housekeeping” genes are continuously highly expressed, so their role in any disease is negligible. We found some “housekeeping” genes contained in the DNBs we obtained, such as GADPH, Ldha, and Arhgdia [18]. These genes are not considered in the subsequent analysis [18].

For every dataset, we use two kinds of GO backgrounds for analysis, one of which includes all *Rattus norvegicus* genes, and the other is only the genes that pass the previous filtering steps (the genes in the NDM). We show the two results of GSE13268 in Table 1 and Table 2, which are analyzed separately. For the GSE13269 and GSE13270 datasets, we only show the results for genes that passed the previous filtering steps as background in Table 3. The results for all genes in GSE13269 and GSE13270 are in Appendix A.

#### 3.2.1. Analysis of GSE13268

For GK rat adipose, the enrichment results of DNBs based on all genes of the species are provided in Table 1. The biological processes at 8 weeks mainly include response to lipid, stimulation, and insulin secretion regulation. Insulin resistance and inflammation are linked with T2DM and related diseases. Abnormal insulin secretion causes hyperglycemia, which will stimulate the development of hyperinsulinemia, reduce the number of insulin receptors, and aggravate insulin resistance [19]. Moreover, insulin resistance in T2DM is accompanied by the dysfunction of glucose and lipid metabolism as well as protein biosynthesis. Therefore, the related biological processes of DNB genes on T2DM also include the regulation of protein phosphorylation and the response to peptide hormones. For cellular calcium ion homeostasis, calcium ion participates as an activator of lipid metabolism [2]. For regulation of the erk1 and erk2 cascade, the increase in DPP-4 content in diabetic mice specifically activates the ERK1/2 and NF-κB signaling pathways and promotes the calcification of their aortic vessels [20]. Renal artery calcification can enhance progressive renal damage in rats with T2DM nephropathy, further leading to hyperlipidemia [21,22].

For some DNB genes, NOS3 polymorphisms are associated with the progression of kidney and cardiovascular disease in Type 2 diabetic patients. FABP1 regulates the absorption and transportation of fatty acids in the liver by promoting the transportation, storage, and utilization of fatty acids and their acyl-CoA derivatives. The overexpression of FABP1 can disrupt the clearance function in autophagy by inhibiting lysosome functions (including lysosomal protein decomposition and lysosomal acidification maintenance), which promotes liver steatosis [23]. The enzyme subunit expressed by the mitochondrial ND3 gene is an important part of respiratory chain complex I. Deficiency of the ND3 gene in β cells will lead to decreased respiratory chain complex I activity, which will result in islet β cell dysfunction and allow for the possibility of developing diabetes [24].

The enrichment results for DNBs based on the genes that pass the previous filtering steps as background are presented in Table 2. The enrichment pathways represented by DNBs mainly include response to lipopolysaccharide, protein kinase binding, endopeptidase activity, oxidation–reduction process, and response to insulin. Statistically significant enrichment is observed in all cases, with *p* values less than 0.05. The relationship between abnormal insulin secretion and T2DM has been explained in previous analyses. Here, the results for response to lipopolysaccharide and protein kinase binding are the same as in the above analysis [19]. In addition, we also obtained new pathways, such as inflammatory response. Either in tissue-specific analysis or in cross-tissue analysis, lipid metabolism that appears to be abnormal or inflammatory is a common functional cascade associated with T2DM [2].

Comparing the analysis results against two different backgrounds, it can be seen that although there are some differences in the pathways identified through enrichment, the types of pathways are basically the same. The DNBs of adipose tissue are sensitive to metabolic processes and lipid responses.

By setting two different backgrounds for analysis, both demonstrate that the DNBs obtained by DMNE are closely related to the development of T2DM.

#### 3.2.2. Analysis of GSE13269

For GK rat muscle, the related biological processes at 4 weeks mainly include muscle contraction, glycolytic process, peptidase activator activity, response to glucose, T cell co-stimulation, cellular response to glucose stimulus, glucose homeostasis, and the PI3K/Akt signaling pathway, which are provided in Table 3. The biological processes at 16 weeks mainly include regulation of peptidase activity, negative regulation of phosphorylation, carbohydrate metabolic process, glycolytic process, T cell co-stimulation, regulation of protein catabolic process, the HIF-1 signaling pathway, and the PI3K/Akt signaling pathway, which are also provided in Table 3. B/T cells actively participate in the inflammatory response in the first critical period for muscle tissues [25,26]. For the PI3K/Akt signaling pathway, the gene Sirt1 participates in diabetic myocardial injury and the occurrence and development of early diabetic cardiomyopathy by negatively regulating the PI3K/Akt/MTOR signaling pathway [27]. Through cell transfection and other experiments, previous studies have shown that the HIF-1α/KIM1 signaling pathway can participate in the process of renal fibrosis in diabetic nephropathy by regulating the KIM1 expression in renal tubular epithelial cells under a high-glucose environment. HIF-1α is the key transcription factor of oxygen homeostasis regulation, which can make the body adapt to the external environment oxygen by regulating the expression of target genes. The expression levels of HIF-1α, KIM1, and COL-1mRNA and protein controlled by them in the kidneys of diabetic rats are significantly increased, suggesting that the diabetic nephropathy model is accompanied by changes in the HIF-1α/KIM1 pathway in kidneys [28]. For some DNB genes, PRDX2 has been found to participate in the oxidative stress process through a variety of signaling pathways. The environment of hypoxia that it induces leads to an increase in reactive oxygen species. In order to survive in a high level of reactive oxygen species, cells need to increase antioxidant levels, so the expression of related factors, including PRDX2 protein, is altered. Previous studies suggest that PRDX2 overexpression can prevent pancreatic cell apoptosis induced by oxidative stress and reduce the risk of developing diabetes [29]. Some studies [30,31] have suggested that SPARC is an autocrine and/or paracrine factor of adipose tissue that can inhibit fat formation. Other studies [32] have shown that insulin can increase the expression of SPARC, which can increase the phosphorylation of AKT and PI3-K, indicating that SPARC may be involved in insulin signaling pathways [33].

#### 3.2.3. Analysis of GSE13270

For GK rat liver, the biological processes at 4 weeks mainly include inflammatory response, response to hormone, oxidoreductase activity, catalytic activity, response to interleukin-1, response to lipid, and metabolism of xenobiotics by cytochrome P450 and the AGE/RAGE signaling pathway in diabetic complications, which are provided in Table 3. The biological processes at 16 weeks mainly include liver development, lipid binding, negative regulation of immune response, the NOS2-CD74 complex, vitamin A metabolic process, the PPAR signaling pathway, and the AGE/RAGE signaling pathway in diabetic complications, which are also provided in Table 3. In the first critical state, the metabolism of xenobiotics by the cytochrome P450 pathway plays an important role in the oxidation of organic substances, whose involved enzymes in liver could generally act as metabolic intermediates, e.g., lipids and steroid hormones [34,35]. Vitamin A and its metabolites can inhibit the expression of resting and activated pancreatic stellate cells α-SMA [36]. The peroxisome proliferator-activated receptors (PPARs) modulate several biological processes that are perturbed in obesity, including inflammation, lipid and glucose metabolism, and overall energy homeostasis. PPARs regulate the functions of adipose tissues, such as adipogenesis, lipid storage, and adaptive thermogenesis [37]. For the AGE/RAGE signaling pathway, moderate-intensity aerobic exercise inhibits the AGE/RAGE axis and the NF-κB pathway, which may decrease oxidative stress and inflammation and thus reduce tissue injury for the prevention and treatment of T2DM complications [38]. For some DNB genes, CD74 receptor and other mechanisms are stimulated by multi-effect cytokine MIF to promote inflammatory response in glomerular podocytes. MIF can activate CD74 on the surface of local glomerular podocytes, resulting in the phosphorylation of extracellular signal-regulated kinase 1/2 (ERK1/2) and p38MAPK [39]. SPARC is an autocrine and paracrine factor of adipose tissue that can inhibit adipogenesis. Insulin can increase SPARC expression, while SPARC can increase Akt and PI3-K phosphorylation, indicating that SPARC may be involved in insulin signaling pathways [40]. PRDX2 overexpression can prevent pancreatic β cell apoptosis induced by oxidative stress and reduce the risk of developing diabetes. The changes in oxidative stress in diabetic kidney disease and other pathological processes are related to the expression of PRDX2 and related pathways. For example, PRDX2 can participate in the regulation of tumor development and oxidative stress in treatment through the PI3/AKT-resistant pathway. PRDX2 can regulate the oxidative stress of colon cancer cells through the Wnt/B-catenin signaling pathway and influence the oxidative stress of tumor by regulating some microRNAs [29].

### 3.3. Cross-Tissue Analysis

From the above enrichment analysis, we observe the critical point of tissue specificity. There are different degrees of differences in the ways DNBs participate in T2DM, but there is also relative consistency. We can infer that the phenotypic changes in GK rats share a cross-organizational functional relationship during the development of T2DM.

In the pre-disease period, the high blood glucose is observed in GK rats fed with the same food in the healthy control WKY rats, although the insulin secretion level in GK rats is similar to the level in WKY rats. Hence, it can be inferred that the insulin receptors in GK rats could not correctly respond to the insulin regulation, such that the glucose cannot be transported successfully in cells. In the preceding analysis, we find that there are abnormal responses to lipids in the early warning stage of lipid tissue and the first early warning stage of liver tissue, and abnormal responses of the PPAR signaling pathway are found in the fourth early warning stage of liver tissue. This pathway is activated by upstream FABP family proteins and is then involved in fatty acid transport. This may explain the remarkable abnormal lipid metabolism in the early stage of the disease due to a lack of glucose intake.

In the pre-transition period, hyperglycemia remains comparatively stable, but the concentration of plasma insulin decreases. The ability to absorb and utilize glucose is reduced. Consequently, other metabolic processes in tissues, such as lipid metabolism and protein metabolism, are activated. Many genes related to lipid transport are dysfunctional in the liver, where disorders in protein digestion and absorption can first be detected. The occurrence of β cell failure at 20 weeks may result from the increasingly serious lack of glucose uptake. The abundance of enrichment pathways in various tissues indicates that with the further development of the disease, complications will increase, and functional proteins will encounter obvious obstacles in their operation, such as in the examples of negative regulation of proteoglycan and regulation of phosphorylation in cancer. Drug metabolism and vitamin metabolism are also abnormal. In the above analysis, it can also be found that the response to stimulation appears throughout the entire development process of T2DM.

### 3.4. DMNE Reveals Non-Differential “Dark Genes”

In clinical practice and scientific research, differentially expressed genes draw much attention in the early diagnosis of disease, screening drug targets, treating diseases, and developing new drugs. However, some non-differentially expressed genes in the coding region of DNA are called “dark genes” [41]. Based on the DMNE method, we found some “dark genes” that did not exhibit differential expression but whose DMNE scores were especially sensitive. Such genes are usually ignored in traditional analyses.

Many epidemiological studies have found that T2DM is positively associated with an increased risk of PAAD. Long-term Type 2 diabetes increases the risk of PAAD by 1.5 to 2 times. Thirty to forty percent of pancreatic cancer patients have diabetes, and eighty percent have abnormal glucose tolerance [42]. Therefore, some “dark genes” related to the prognosis of PAAD may play important roles in the development of T2DM, such as COL1A1, which is the most significant gene in the extracellular matrix receptor interaction pathway and is linked to hypoglycemic activity [43], which may affect blood glucose fluctuations, T2DM prognosis, and the occurrence and development of chronic complications [44].

To further confirm the effectiveness of the “dark genes”, we subjected them to GO and KEGG enrichments analyses. BCKDHA exists in liver tissue and encodes the BCKDC component. In the study of [45], it was found that knocking out the BCKDK gene, which regulates BCKDC kinase, can significantly reduce the concentration of branched-chain amino acid (BCAA) in blood. BCAA can stimulate mammalian targets of rapamycin and S6 kinase. It can also phosphorylate insulin receptor substrate on serine residues, thus damaging insulin signaling pathways. The decrease in BCAA catabolism will lead to insulin resistance and glucose intolerance, eventually leading to T2DM. Therefore, the increase in BCKDHA expression will cause a decrease in plasma BCAA concentration, subsequently affecting the progression of T2DM, which can explain why the low BCKDHA expression is associated with favorable prognosis [43]. COL1A1 contributes to T2DM via the ECM–receptor interaction pathway. ECM–receptor interactions are microenvironmental regulators of the structure and function of cells and tissues. ECM synthesis is closely linked to the PI3K/Akt pathway and the regulation of T2DM [46,47]. The enrichment analysis of these “dark genes” also showed that their associated pathways provided in Figure 4 are related to T2DM. Figure 4 shows the enrichment pathways of “dark genes” and the number of involved genes.

As pancreatic adenocarcinoma (PAAD) and T2DM are interrelated, and this association could inform our understanding of the mechanisms of both diseases, this relationship has become a topic of interest receiving considerable attention in the literature [42,48,49,50]. At present, due to the lack of corresponding prognostic data in the study of T2DM, the prognostic analysis of T2DM cannot be directly performed. Therefore, we used the PAAD data to analyze the prognosis of T2DM, as detailed in Appendix A, representing an attempt to reveal the mechanism of T2DM development from another approach.

### 3.5. Drug Targets

Identifying drug targets is a key task in drug discovery and chemical genomics. In this study, we collected information on the targets and anticancer drugs for T2DM from the online service web pages Drug Bank, Therapeutic Target Database, and Pharmacogenomics Knowledgebase. The number of drug targets in the first, second, third, fourth, and fifth stages of the data of GSE13268 is 20, 29, 21, 23, and 19, respectively. The number of drug targets in the first, second, third, fourth, and fifth stages of the data of GSE13269 is 17, 20, 19, 20, and 15, respectively. The number of drug targets in the first, second, third, fourth, and fifth stages of the data of GSE13270 is 20, 35, 33, 37, and 29, respectively. We integrated the drug target information for critical states (phase 2 of GSE13268, phases 1 and 4 of GSE13269, and phases 1 and 4 of GSE13270).

The critical state of GSE13268 is the second stage, the critical states of GSE13269 are the first and fourth stages, and the critical states of GSE13270 are the first and fourth stages. Figure 5 shows the drug target analysis for the critical states of GSE13270. The drug target analyses [51,52,53,54,55] of GSE13268 and GSE13269 are provided in Appendix A.

In Figure 5, the red circle is the abbreviation of Type 2 diabetes, the blue circle is the drug target of the corresponding stage in the three sets of data of Type 2 diabetes, the green circle is the corresponding target protein, and the yellow circle is the corresponding anticancer drug of the protein that has been approved by the FDA or used in experiments. Infographics of drug targets for other stages of GSE13270 are provided in Appendix A.

In the first stage of GSE13270, according to the analysis of MaoA and MaoB protein expression in the islets of Type 2 diabetes mellitus in mice and humans, it can be found that the amount of MaoB in β cells of Type 2 diabetes mellitus is significantly reduced, which shows that the loss of Mao causes dysfunction of β cells, resulting in Type 2 diabetes [56]. In other studies, a relationship between LGALS2 and glucose and fasting insulin has also been found. It is this relationship that supports the idea that LTA plays a role in determining the insulin–glucose spectrum. LGALS2 genotypes and LTA pathways may play other roles in insulin metabolism which require further research and exploration [57]. In the fourth stage of GSE13270 (liver tissue), FABP2, the intestinal fatty-acid-binding protein, is considered a candidate gene for diabetes. The reason is that, in the process of the absorption and metabolism of fatty acids (FA), FABP2 is involved in encoding proteins. Therefore, FABP2 may affect insulin sensitivity and glucose metabolism [58]. The gene-encoding alpha2 Heremans Schmid glycoprotein (AHSG) is a candidate for Type 2 diabetes and metabolic syndrome, and variants have been successfully demonstrated in previous attempts to be associated with Type 2 diabetes and obesity when examining Swedish patients, as well as Caucasian patients in France [59].

## 4. Discussion

In this study, we propose a new method, the DMNE algorithm, to detect the critical state and identify DNBs of T2DM. By applying this method to three rat gene expression datasets involving adipose tissue, gastrocnemius muscle, and liver tissue in healthy control vs. Type 2 diabetes conditions, the early warning signals of T2DM were successfully detected and DNBs were effectively identified. Specifically, for the three tissues, the DMNE method can not only detect the critical state via the DNBs but also identify drug targets. We used two kinds of GO backgrounds for analysis, and both demonstrated that the DNBs obtained by DMNE are closely linked to the development of T2DM. In addition, we mined “dark genes” that are associated with T2DM. In contrast to the information of differential expression used in traditional biomarkers to diagnose disease, DMNE is based on molecular network fluctuation and is able to predict disease.

We detected the critical state by exploiting the high-dimensional information of time series data. Firstly, by constructing the SSN, we converted floating gene expression into a stable SSN. Then, we calculated the local degree matrix network entropy and degree matrix network entropy according to the information of the differential network. The DMNE method reliably quantifies network fluctuation, i.e., collective fluctuation of molecules caused by a perturbed sample against a group of given reference samples, so as to reduce the noise and thus enhance the robustness and effectiveness by exploring dynamical and high-dimensional information of omics data. Based on the DMNE method, we can detect the critical states before the disease occurs and identify the DNBs. Furthermore, based on the DMNE method, “dark genes” related to disease progression can be identified. Although those “dark genes” are not differentially expressed genes, they are associated with disease progression in T2DM patients and were validated by the functional analysis.

Furthermore, it is noteworthy that the degree matrix network entropy (DMNE) is based on the dynamical network biomarker (DNB), and it can detect the critical states by calculating the score of each stage and using the difference in scores to show the changes in the development of diseases. Therefore, it can be directly applied to stage-course disease datasets in theory, such as stage-course tumor datasets and diabetes datasets, only requiring the modification of corresponding parameters based on sample data and the number of genes when constructing the network. However, it may not be possible to detect the critical states for the disease with fewer than three stages, such as the datasets divided into reference and control only but not staged, because DNB theory for early-warning is not applicable to diseases with fewer than three stages. If a dataset includes fewer than three stages, no valid score comparisons can be made.

In summary, we proposed a robust and effective algorithm that can be used not only to detect critical states of T2DM development and identify DNBs by using stage-course data but also to identify the important “dark genes” and pinpoint drug targets.

## Figures and Tables

**Figure 1 entropy-24-01249-f001:**
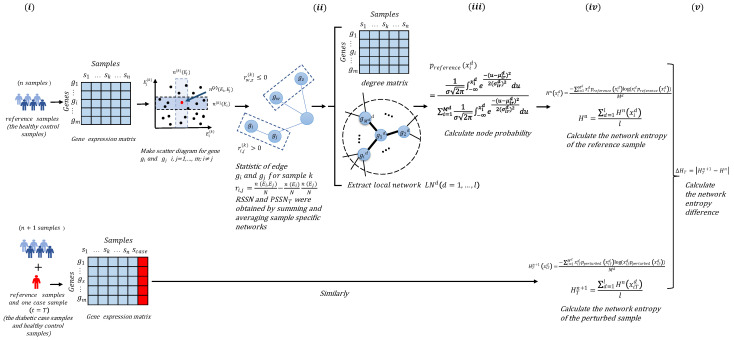
Schematic illustration of the degree matrix network entropy (DMNE) algorithm. (**i**) Construct the SSN of reference samples (the healthy control samples) and perturbed samples (the diabetic samples), respectively. (**ii**) Extract degree matrix and local network. (**iii**) Calculate the node probability for every gene by fitting a Gaussian distribution. (**iv**) Calculate the network entropy of reference samples and perturbed samples. (**v**) Calculate the differential network entropy based on the degree matrix to obtain DMNE score.

**Figure 2 entropy-24-01249-f002:**
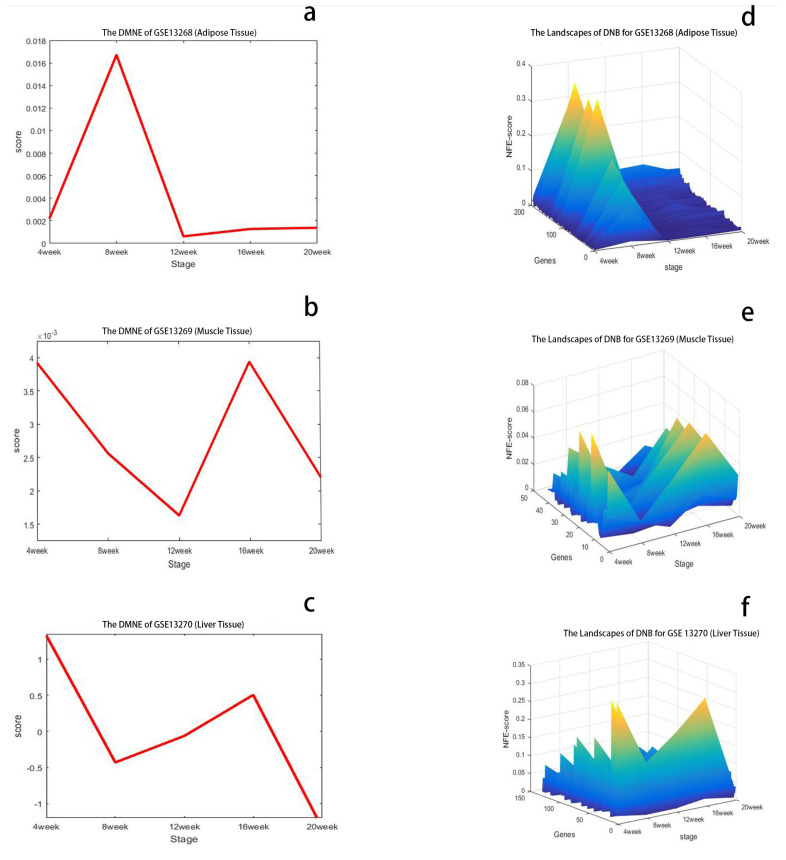
Identification of the critical states of T2DM. (**a**) DMNE score curve of GSE13268, for rat adipose tissue, during T2DM, which shows the critical state around 8 weeks. (**b**) DMNE score curve of GSE13269, for rat gastrocnemius muscle tissue, during T2DM, which shows the critical states around 4 and 16 weeks. (**c**) DMNE score curve of GSE13270, for rat liver tissue, during T2DM, which shows the critical states around 4 and 16 weeks. (**d**–**f**) The dynamic changes in degree matrix network entropy (DMNE) scores demonstrate the landscape of network entropy. (**g**–**i**) Dynamic evolution of dynamic network biomarkers (DNBs) of three tissues for T2DM, where the red nodes represent DNBs and the blue nodes represent non-DNBs. In (**h**), it can be seen that the DNBs are more active and the network structure undergoes significant changes at 4 and 16 weeks. By using the DMNE approach, the early warning signals before the critical transition into disease state can be detected by taking a network perspective.

**Figure 3 entropy-24-01249-f003:**
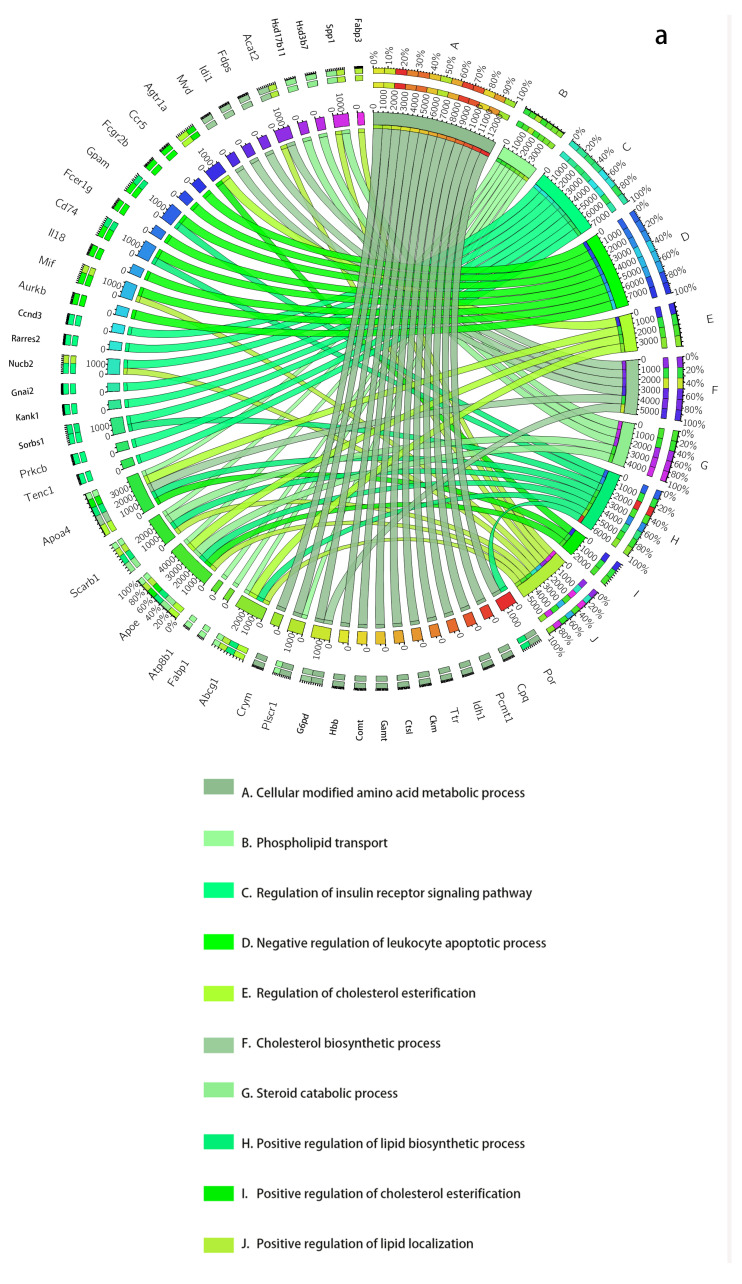
The dynamic network biomarkers (DNBs) of GSE13268, involving rat adipose tissue, are involved in important biological processes of T2DM. (**a**) All *Rattus norvegicus* genes as background. (**b**) The genes that passed the previous filtering steps as background. The left side of the outer ring represents the detected DNB members, and the right side represents detailed biological processes in which these genes are involved. In the inner ring, the color and width of links indicate diverse enrichment pathways and significant levels of gene functions, respectively.

**Figure 4 entropy-24-01249-f004:**
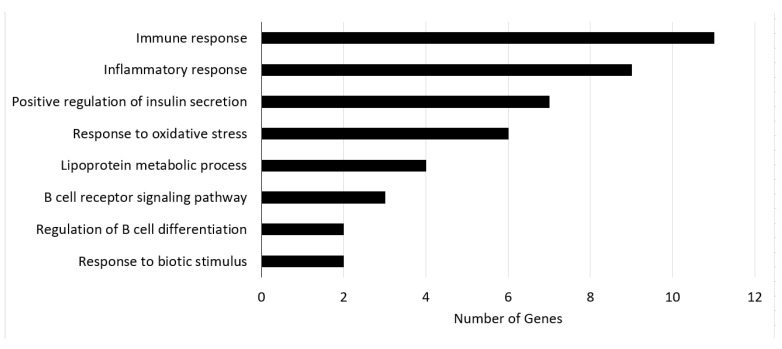
The enrichment pathways of “dark genes” and the number of involved genes.

**Figure 5 entropy-24-01249-f005:**
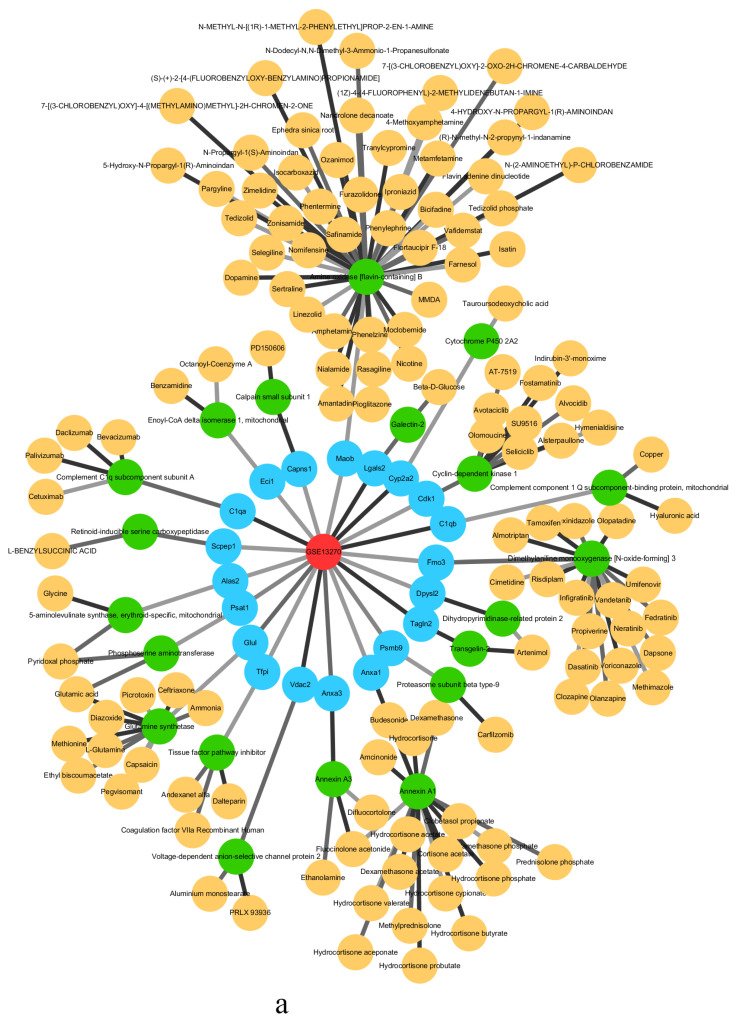
The drug target analysis in GEO study GSE13270 (rat liver tissue) showing the drug targets and corresponding protein and targeted drugs at (**a**) stage I and (**b**) stage IV.

**Table 1 entropy-24-01249-t001:** Enrichment results for DNBs based on all genes of the species in GSE13268 (adipose tissue).

Tissue	Case	Term	*p*-Value	Term Name
Adipose	Adipose 8 weeks	GO:0033993	3.37 × 10^−9^	Response to lipid
		GO:0050896	1.33 × 10^−8^	Response to stimulus
		GO:0048583	2.51 × 10^−5^	Regulation of response to stimulus
		GO:0006629	0.0238	Lipid metabolic process
		GO:0050796	0.0238	Regulation of insulin secretion
		GO:0001932	0.0049	Regulation of protein phosphorylation
		GO:0043434	0.0124	Response to peptide hormone
		GO:0070372	0.0159	Regulation of erk1 and erk2 cascade
		GO:0006874	0.0235	Cellular calcium ion homeostasis

**Table 2 entropy-24-01249-t002:** Enrichment results for DNBs based on the genes that passed the previous filtering steps in GSE13268 (adipose tissue).

Tissue	Case	Term	*p*-Value	Term Name
Adipose	Adipose 8 weeks	GO:0032496	0.004939298	Response to lipopolysaccharide
		GO:0019901	0.008557741	Protein kinase binding
		GO:0004175	0.012946747	Endopeptidase activity
		GO:0055114	0.013323312	Oxidation–reduction process
		GO:0032868	0.016048825	Response to insulin
		GO:0033700	0.01966288	Phospholipid efflux
		GO:0051384	0.020031683	Response to glucocorticoid
		GO:0048545	0.031308927	Response to steroid hormone
		GO:0006954	0.031444423	Inflammatory response
		GO:0008203	0.035288946	Positive regulation of B cell receptor signaling pathway
		GO:0055088	0.03504665	Lipid homeostasis
		GO:0050729	0.038857846	Positive regulation of inflammatory response
		GO:0008203	0.047115437	Cholesterol metabolic process

**Table 3 entropy-24-01249-t003:** Enrichment results for DNBs of GSE13269 (gastrocnemius muscle tissue) and GSE13270 (liver tissue) datasets.

Tissue	Case	Term	*p*-Value	Term Name
Muscle	Muscle 4 weeks	GO:0006936	0.001436732	Muscle contraction
		GO:0006096	0.005358956	Glycolytic process
		GO:0016504	0.011881481	Peptidase activator activity
		GO:0009749	0.013277291	Response to glucose
		GO:0031295	0.019842374	T cell co-stimulation
		GO:0071333	0.03156758	Cellular response to glucose stimulus
		GO:0042593	0.048961141	Glucose homeostasis
		rno00190	0.019507611	PI3K-Akt signaling pathway
	Muscle 16 weeks	GO:0052547	0.019261847	Regulation of peptidase activity
		GO:0042326	0.021460151	Negative regulation of phosphorylation
		GO:0005975	0.017537912	Carbohydrate metabolic process
		GO:0006096	0.000947641	Glycolytic process
		GO:0031295	0.02116544	T cell co-stimulation
		GO:0042176	0.037110591	Regulation of protein catabolic process
		rno04066	0.007510911	HIF-1 signaling pathway
		rno04151	0.006275621	PI3K-Akt signaling pathway
Liver	Liver 4 weeks	GO:0006954	0.001396509	Inflammatory response
		GO:0009725	0.002627081	Response to hormone
		GO:0016491	0.002701852	Oxidoreductase activity
		GO:0003824	0.013755671	Catalytic activity
		GO:0070555	0.014842206	Response to interleukin-1
		GO:0033993	0.027356508	Response to lipid
		rno00980	0.017258367	Metabolism of xenobiotics by cytochrome P450
		rno04933	0.027963125	AGE-RAGE signaling pathway in diabetic complications
	Liver 16 weeks	GO:0001889	0.00703339	Liver development
		GO:0008289	0.00291932	Lipid binding
		GO:0050777	0.005409903	Negative regulation of immune response
		GO:0035693	0.014337447	NOS2-CD74 complex
		GO:0006776	0.028470377	Vitamin A metabolic process
		rno03320	0.003125368	PPAR signaling pathway
		rno04933	0.006912547	AGE-RAGE signaling pathway in diabetic complications
		rno00982	0.008912531	Drug metabolism—cytochrome P450

## Data Availability

The high-throughput experimental datasets were downloaded from the NCBI GEO database (access ID: GSE13268, GSE13269 and GSE13270) (www.ncbi.nlm.nih. gov/geo/ (accessed on 1 October 2021)). Pancreatic adenocarcinoma (PAAD) survival data and the gene expression values in different samples were downloaded from The Cancer Genome Atlas (TCGA) database (https://cancerge nome.nih.gov/ (accessed on 1 March 2022)). The drug targets were downloaded from DrugBank (https://go.drugbank.com/ (accessed on 20 January 2022)), Therapeutic Target Database (TTD, http://db.idrblab.net/ttd/ (accessed on 20 January 2022)), and Pharmacogenomics Knowledgebase (PharmGKB, https://www.pharmgkb.org/ (accessed on 20 January 2022)). The original codes are available at https://github.com/yyk124/DMNE-main.git (accessed on 26 July 2022).

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
