# Peer review of "Detecting the Critical States of Type 2 Diabetes Mellitus Based on Degree Matrix Network Entropy by Cross-Tissue Analysis"

_entropy, 2022, doi:10.3390/e24091249_

Round 1
Reviewer 1 Report
Yang et al used gene expression data to construct a network and then calculate the entropy compared to a reference network to find the hidden genes in developing type 2 diabetes. I have issues with the overall concept and the description of the data processing. Due to that, the study needs to be redone and the results reevaluated.
Major:
1. The choice of network generation is my most significant problem. The authors described those genes are interacting which have a higher then chance of having the same expression based on a threshold -if I got the scatter plot correctly. I disagree with this. (1) the gene expression of interacting genes should not be the same in the biological system -housekeeping genes have high expression. In the manuscript, the authors did not mention that they accounted for that. (2) The described method uses arbitrary cut-offs (0.1*k in the paper) a much more sophisticated method would be a correlation if the authors want to build a gene correction network.
2. The description of normalisation and handling of the gene expression data is insufficient. The authors need to describe what packages they used in what kind of environment (R, python). What steps do the authors after downloading the data? Did they use the available preprocessed gene expression data? How have the authors used DAVID? What was the background of gene ontology analysis?
3. The authors moved from type 2 diabetes to pancreatic cancer. It is a different disease. There is a correlation between type 2 diabetes and pancreatic cancer, but I disagree to use pancreatic cancer as an example to check the dark genes from type two diabetes mellitus as prognostic factors. I suggest using them to correlate with the pathomechanism of type 2 diabetes instead only. Also, the description of using the pancreatic cancer data set is missing from the methods.
Minor:
1. Please cite the used methods, databases' papers as well not just the website.
2. Have the authors filtered STRING? What was it used for?
3. Figure 1 is hard to follow. I suggest not writing down the whole Gaussian function in every part and following it from left tonight without a step-down and then up.
4. Figure 2 GHI are unclear. What do the authors want to say? I see red dots and it is not clear to me.
5. Figure 3 The circos plots are too small the legend is not legible.
6. In Figure 6 the drug names are not legible.
I loved the idea of using gene expression to uncover the hidden mechanism of a complex disease but in the current form, I have to suggest rejecting the manuscript. The usage of the gene expression data is not described and sadly that by itself invalidates any further, hovewer interesting conclusions.
Sincerely yours,
Dezso Modos
Reviewer 2 Report
Reviewer’s Comments
Summary and general comments
In this manuscript, the authors investigated the significance of detecting the critical states of type 2 diabetes mellitus by using adipose, muscle, and liver tissue development based on degree matrix network entropy. The directions are sound and interesting. However, the results presented are not convincing to support the authors’ conclusions. In addition, the classification performed with the degree matrix network entropy did not provide satisfactory decision functions.
Major comments
1. The authors need to change the title. Make sure what is originality! system design? algorithms? clinical applications?
2. “By applying the method to the datasets of three different tissue samples of T2DM, the critical states are detected and the dynamic network biomarkers are identified successfully.” in abstract needs to be clearly presented. The sentence “In addition, we found some ‘dark genes’, and identified some drug targets.” in the abstract is unexpected.
3. In the manuscript, the different groups are not adequately argued in the discussion. The authors need to investigate the grouping.
4. The authors need to clarify whether the testing rats had hypertension and/or arrhythmia. In addition, the authors should mention whether the rats were under medical treatment which may affect the results at the time of the study. These points need to be addressed.
5. The statistical analysis is not performed appropriately and rigorously. The p values in the tables need to be addressed in detail. The authors could use *p < 0.05 Group 1 vs. Group 2, **p< 0.001 Group 1 vs. Group 2. A p-value <0.05 was noted as statistically significant. Subsequently, the normality distribution and the homoscedasticity of variables have to be checked.
Minor comments
1. The statement in lines 77-78 is not appropriate for the end of the introduction.
2. Abstract should be more precise.
3. Eq.(2) and Eq.(3) are not consistent, please check?
4. The Figures were not clear for publication.
Round 2
Reviewer 1 Report
The paper is getting there the authors excluded the pancreas cancer diabetes line, which I really appreciate and tried to include the description of the gene expression preprocessing, but that part needs further work. The figures need to be larger to be legible.
I will answer the author's detailed message.
Major Question 1:
Point 1 The authors misunderstood me. Highly expressed genes are false positives in the work. Housekeeping genes are continuously highly expressed so their role in any disease is negligible. Please explain in the article how specific is the proposed method?
Point 2 I absolutely accept the author's response, but please write it into the manuscript as well.
Question 2:
Points 1-2 The description of data and preprocessing is better but still insufficient. Please describe the normalisation methods. The authors mentioned that they used the raw data. However, they did not mention any commonly used packages which are used in microarray preprocessing. The input for the method is log2 normalised gene expression values, at least that is what they have in the GitHub folder of the DMNE method (https://github.com/yyk124/DMNE-main/blob/main/DMNE_method.zip). Please provide the steps on how they get there, also the scripts. If the authors downloaded the gene expression matrix files from GEO those data are not raw data. They are preprocessed data. It is ok to use that, but please state that in the answer and in the paper.
Also, the authors excluded those probesets which were mapped to multiple genes. That can bias the outcomes.
Point 3: The authors made a mistake in Gene Ontology Analysis. The background universe of the genes had included the genes which they excluded from their analysis (genes which did not have expression, genes which were represented by probesets mapping to multiple genes). This by itself introduce bias in the hypergeometric tests which DAVID uses. Ideally, when using DAVID it is preferable to use the background in which we have information. In this case, I suggest using the genes which passed the previous filtering steps.
Question 3:
Thank you very much appreciated.
Minor:
Question 1: Thanks approved
Question 2: Please describe the methods as well.
Question 3: Thanks
Question 4: The figure is still meaningless. I guess the hairballs are around 15 k genes. Please see https://journals.plos.org/ploscompbiol/article?id=10.1371/journal.pcbi.1007244 for network visualisation.
Question 5: No it is still not visible with 400% magnification due to the pdf's jpeg compression. Please make the figures bigger - 1 page large three circos plot.
Question 6: Same problems as in Questions 4 and 5. It is hard to understand what the network figure communicates and sadly due to the pdf conversion the labels are not legible.
Reviewer 2 Report
By considering the major revision of the original manuscript, I am proposing the acceptance of this work. The authors properly addressed the comments and suggestions of the reviewers.
Author Response
It is a great honor to have your recognition of this work.
Round 3
Reviewer 1 Report
The authors addressed my comments. I wish you all the best for further work!
Author Response

(The authors gave the same response as above.)
